# Assessment of Preferences in Taking Painkillers among Students of Medicine, Dentistry, and Pharmacy: A Pilot Study

**DOI:** 10.3390/healthcare12020196

**Published:** 2024-01-13

**Authors:** Wiktoria Samorek, Joanna Przylepa, Joanna Urbaniak, Aleksandra Rogala, Anna Pilimon, Krzysztof Błochowiak, Katarzyna Błochowiak

**Affiliations:** 1Department of Oral Surgery, Periodontal Diseases and Oral Mucosal Diseases, Poznan University of Medical Sciences, 61-701 Poznan, Poland; wiki_1101@wp.pl (W.S.); joannaprzylepa01@wp.pl (J.P.); jo_urba@wp.pl (J.U.); aleksandra.rogala2@wp.pl (A.R.); ania_pilimon@wp.pl (A.P.); 2Medical Faculty, Medical University of Gdańsk, 80-210 Gdańsk, Poland; blochowiakkrzysztof@gumed.edu.pl

**Keywords:** painkillers, medical students, preferences, pain, analgesics, dental students, toothache

## Abstract

Students of pharmacy, medicine, and dentistry are important for shaping drug policy. The aim of this study is to assess and compare students preferences in taking painkillers. The study group consists of 382 students of pharmacy (28.8%), medicine (40.0%), and dentistry (30.1%). An anonymous questionnaire consisting of 17 questions was prepared using the Google Forms platform and distributed through social media. Ibuprofen was the most frequently preferred, regardless of the study major (57.8%). Pharmacy students expressed the least concern about the possible side effects of analgesics (17.5%). The fast onset of painkillers was more important for dental students (59.1%) and pharmacy students (44.7%), compared to medical students (39.22%). Medicine and pharmacy students indicated their studies to be their main source of information about painkillers compared to dentistry students (*p* = 0.001). There are no differences in pain severity regarding which analgesics are used among student groups (*p* = 0.547). Dental students experienced odontogenic pain less frequently (57.3%) than medical (79.7%) and pharmacy students (79.8%), (*p* = 0.000). Ketoprofen was the most frequently chosen prescription painkiller for odontogenic pain in all groups (49.4%). Gastrointestinal complaints were the most often reported side effects, regardless of the study major (87.1%). Choice of studies, gender, and year of study were the most important determinants of the choice of painkillers.

## 1. Introduction

Pain is a complex sensory and emotional experience in response to actual or potential tissue damage. Its intensity depends on individual biological and psychological characteristics. The development and transformation of acute and chronic pain can result from the initial experience of pain and individual biopsychosocial factors. Pain is a common experience and occurs in all people, regardless of age. Previous studies on pain occurrence showed that, during a three-month period, 29% of adults experienced low back pain, 17% of them reported migraine or severe headache, 15% of adults experienced neck pain, and 5% complained of facial or jaw pain, including acute odontogenic pain [1]. The prevalence of experiencing pain has led to the search for methods to effectively combat it. Apart from identifying and eliminating the cause of pain and applying disease-specific treatment, a recommended management is the use of analgesics. Previous studies revealed that analgesics are the most commonly continued or newly prescribed medication on an ambulatory basis, and this prevalence is estimated at 11.4% [1]. Many analgesic treatment regimens have also been introduced, depending on the location of pain, its nature, and its intensity, as well as taking into account the contraindications, side effects, and precautions for selected analgesics. The regimens include several groups of analgesics and adjuvant drugs [1,2]. Many over-the-counter painkillers are commonly used to relieve mild to moderate pain. One of the most common types of acute and non-malignant pain is odontogenic pain. Preferences in the choice of painkillers among the general population are based on various, often not previously identified, reasons; however, these do not include specialist data on the safety of their use, contraindications, and side effects [3]. Popular knowledge regarding analgesics is limited and built up based on personal experience, advice from doctors, pharmacists, other patients, or even TV commercials and internet portals [4]. From this point of view, students of pharmacy, medicine, and dentistry are a valuable research group to assess preferences in the choice of painkillers and the formation of these preferences depending on the knowledge of their mechanism of action. Therefore, analyzing preferences in the choice of analgesics among students of medical universities seems important, and may lead to the development of models of shared decision-making that take into account both patients’ and clinicians’ opinions concerning the clinical benefits, adverse effects, preferences, and costs of painkillers. This might improve care and patients’ adherence to prescribed treatment regimens. Understanding the factors that determine their treatment preferences might improve the quality of this shared decision-making process and the overall success of treatment. Therefore, in order to address these needs, we conducted a survey on the preferences of painkiller use among students of dentistry, medicine, and pharmacy. As the research hypothesis, we state that the knowledge acquired during studies, along with students’ character and gender, may determine their choice of painkillers.

The aim of this study is to assess and compare preferences when taking painkillers among dental, medical, and pharmacy students, and to find possible factors determining their choice, such as the price of medicine, the drug’s accessibility, the impact of the advertisement, and gender. An additional aim is to study preferences regarding whether painkillers are effective in reducing pain of dental origin, one of the most common types of severe and non-malignant pain.

## 2. Materials and Methods

### 2.1. Study Groups

A total of 382 students of medicine, dentistry, and pharmacy in all years of education from universities all over Poland participated in the project (male:female ratio 79 (20.6%):303 (79.1%)) at an age ranging from 20 to 25 years. The research was conducted using the Google Forms platform and distributed through social media such as Facebook and Instagram. The study ensured full anonymity for individual students and the universities at which they studied. Participation in the study was voluntary. To assess the required number of respondents among medicine, dentistry, and pharmacy students we used the data from the Central Statistical Office in Poland related to the total number of students of medicine, dentistry, and pharmacy in the academic year 2022/2023 in Poland. To set the needed number of respondents in each study group of students, we used the following values: d = 0.10 (an absolute precision), α = 0.05, N (population size). Therefore, we received the required number of students from medicine, pharmacy and dentistry faculties for the study to be valuable: medical students N = 96, dental students N = 94, pharmacy students N = 95. Ultimately, the study comprised 153 medical students, 115 dental students, and 114 students of pharmacy. It was conducted from January 2023 to February 2023 and consisted of the single and anonymous completion of a proprietary online questionnaire, made available to students of medicine, dentistry, and pharmacy. Adult students of medicine, dentistry, and pharmacy were included in the study. Exclusion criteria were the presence of diseases requiring the application of chronic painkillers and pregnancy.

### 2.2. Survey Design

An anonymous questionnaire consisting of 17 questions, including 3 categorizing questions about gender, field of study, and year of study, was used for the auditorium study. It related, among others, to the personal preferences of students in their choice of painkillers, as well as the choices they would propose to their potential patients (generally healthy, over 18 years of age, and non-pregnant women). Six questions were used to measure the drugs used by students. Some of the questions were open-ended, where the respondent could independently indicate an answer, and the vast majority of questions had several possible answers to choose from. The questions contained in the survey were prepared after the initial interview with the student representatives from the medicine, dentistry, and pharmacy faculties. We tried to collect and include all suggestions reported in the interviews. The exact form of the questionnaire is attached as a Appendix A.

### 2.3. Ethical Issues

This study was performed in accordance with the ethical standards laid down in an appropriate version of the World Medical Association Declaration of Helsinki. According to the recommendations of the Bioethics Committee of the Poznan University of Medical Sciences, completely anonymous surveys based on own questionnaires and not relating to the spheres of intimate life or to psychiatry and pediatrics do not require the consent of the committee. According to the Polish law and GCP regulations, the scientific research entitled “Assessment of preferences in taking painkillers among students of medicine, dentistry and pharmacy” does not require the approval of the Bioethics Committee at Poznan University of Medical Sciences. The Bioethics Committee at Poznan University of Medical Sciences confirmed that this research is not a medical experiment (KB-887/23).

### 2.4. Statistical Analysis

The calculations were carried out with Microsoft Excel 2016 and STATISTICA software (v.13 TIBCO, Palo Alto, CA, USA) and PQStat software (v.1.8.6). Categorical variables are presented in contingency tables, and their associations were tested, depending on the number of cases, with Chi^2^ Pearson’s test or Fisher–Freeman–Halton’s test. For qualitative variables, the numbers (*n*) and proportions (%) were calculated and collected in cross-tables. The post hoc tests for multiple comparisons, using the Benjamini–Hochberg method, were used to compare medical students and dental students, medical students and pharmacy students, and dental students and pharmacy students, as well as students in their ≤3rd year of the study and ≥4th year of the study. *p* < 0.05 was considered statistically significant.

## 3. Results

Summarized student demographic characteristics and summarized answers to questions with no division into fields of study, as well as with division into fields of the study, are presented in Table 1.

The use of painkillers is widespread among all surveyed student groups, but they are most often preferred for severe (47.64%) or moderate pain (28.53%). The vast majority of respondents declared that they take painkillers once a month (47.12%) or less than once a (33.50%). Drugs application more than once a week was rare among the respondents (3.93%). A few students were taking analgesics to prevent pain and in the case of mild pain. Adherence to the recommended doses was observed in a high percentage of respondents, it is as much as 74.83% of respondents. Students not adhering to the recommended doses accounted for only 2.88% of all respondents. For the respondents, the most important property of an analgesic was its fast onset of action (46.86%), followed by its potency of action (32.72%). For the vast majority of the survey participants, the side effects associated with taking painkillers were irrelevant. Gastrointestinal disorders were declared the most common side effects during therapy.

Of the suggested over-the-counter painkillers for acute pain, ibuprofen was the most frequently chosen. A detailed distribution of preferences in the choice of over-the-counter painkillers for acute pain is presented in Figure 1.

### 3.1. Comparison of the Use of Analgesics among Students of Medicine, Dentistry, and Pharmacy

The data obtained in the surveys showed that the use of analgesics is widespread among all the students. There were statistically significant differences in the use of analgesics between the studied groups (*p* = 0.034) (Table 1). The study showed that students of medicine use painkillers less often than students of other faculties. The obtained data show no relationship between the field of study and adherence to the recommended doses of drugs. There are no statistically significant differences in the pain intensity at which analgesics are used among the surveyed groups of students (*p* = 0.547). However, there were statistically significant differences in the choice of analgesics and the factors determining this choice among the study groups (*p* = 0.008). A fast onset of analgesic effect was the most important for 59.1% of dental students, 44.7% of pharmacy students, and for 39.2% of medical students. For medical students, potency turned out to be almost equally important (33.9%). In addition, a long duration of drug action (12.4%) also turned out to be relevant for the surveyed medical students, which makes them stand out from other majors. Drug potency was crucial for (22.6%) of dental students, and for (41.2%) of pharmacy students. There is a statistically significant difference between the surveyed groups of students regarding the side effects of painkillers (*p* = 0.047). Pharmacy students expressed the least concern about possible side effects (17.5%). In turn, students of medicine and dentistry expressed greater concern about the possible side effects of painkillers (27.4%) and (31.3%), respectively. Among the possible side effects of analgesics application, gastrointestinal complaints were the most often reported, regardless of the surveyed group of students. There was a statistically significant difference in the choice of analgesics for acute pain (*p* = 0.027). Although ibuprofen was the most frequently chosen over-the-counter painkiller by all the participants in the study, regardless of the field of study, statistically significant differences were found in the use of paracetamol and metamizole. Medical students were statistically more likely to choose paracetamol and metamizole than dental and pharmacy students. All students, regardless of the field of study, do not apply aspirin to fight acute pain. Contraindications to the use of painkillers were declared by 6.02% of the respondents, regardless of the field of study. Students of all study groups were not motivated by advertisements and commercials when choosing painkillers. In turn, there is a statistically significant difference in the indicated source of information regarding painkillers among students of different majors (*p* = 0.001). More than half of the students of medicine and pharmacy indicated their study curriculum to be their main source of knowledge about painkillers. However, only 31.3% of dental students declared this to be a source of knowledge. Among dental students, a relatively large group, compared to students of medicine and pharmacy, obtain information about painkillers from doctors and pharmacists (13.9%), as well as from friends and family (14.7%). Gaining information on painkillers from a drug leaflet and from the Internet was indicated by a comparable percentage of students of medicine, dentistry and pharmacy (Figure 2).

There is a statistically significant relationship between the field of study and experiencing toothache (*p* = 0.000). Statistically significantly lower percentage of dental students did not experience tooth-related pain as compared to the students of medicine or pharmacy (Figure 3).

There were no statistically significant differences in the choice of over-the-counter painkillers to fight toothache among students in each of the studied majors (*p* = 0.805). The drug of first choice for toothache was ibuprofen in all study groups. Dental pain would be relieved by 4.3% of dental students, 9.8% of medical students, and 8.7% of pharmacy students with a prescription drug. However, 6.9% of dental students, 8.8% of medical students, and 4.3% of pharmacy students would not recommend any over-the-counter painkillers for dental pain. On the other hand, there was a statistically significant difference in the choice of prescription analgesic for odontogenic pain (*p* = 0.032). Ketoprofen was the most frequently chosen drug in all study groups. Another frequently chosen drug among students of dentistry was nimesulide, which was not popular among students of medicine and pharmacy.

In post hoc tests for multiple comparisons using the Benjamini–Hochberg method to compare dental students to medical students, there was a statistically significant difference between a fast onset of action and the analgesic drug’s potency and duration of analgesic effect (*p* = 0.042). In the same comparison for dental students and medical students, there was a statistically significant difference between ibuprofen and metamizole in terms of their use in cases of acute pain (*p* = 0.026). Comparing dental students to pharmacy students, there was a difference between the study curriculum and family and friends being indicated as a way to obtain information about individual painkillers (*p* = 0.002). There was a statistically significant difference between toothache experience between dental students and both medical students (*p* = 0.000) and pharmacy students (*p* = 0.000).

### 3.2. Relationship between the Year of Study and the Use of Analgesics

A detailed correlation between the answers and the year of study are presented in Table 2. As the answers show, students of later years took painkillers more often than once a week in comparison to the students from earlier years (*p* = 0.016). The students of later years were statistically more likely to declare their studies to be their main source of knowledge about painkillers (*p* = 0.000).

In post hoc tests for multiple comparisons with the Benjamini–Hochberg method, comparing students in their ≤3rd year of the study and ≥4th year of the study, there was a statistically significant difference between the answers “studies curriculum” and “drug leaflet” (*p* = 0.017), and between “studies curriculum” and “doctors and pharmacists” (*p* = 0.005), and between “studies curriculum” and “friends and family” (*p* = 0.000), and between “family and friends” and “the Internet” (*p* = 0.014) as ways of obtaining information about painkillers. There was a statistically significant difference between ibuprofen and the answer “I would not recommend any” as a recommendation for toothache (*p* = 0.044), and between aspirin and “I would not recommend any” (*p* = 0.044). Finally, there was a difference in the prescription drugs recommended for dental pain when comparing Nimesulide and Tramadol (*p* = 0.023), Ketoprofen and Nimesulide (*p* = 0.006), and Celecoxib and Nimesulide (*p* = 0.031).

### 3.3. Correlation between the Field of Study and the Experience of Pain of Dental Origin and Its Control

There is a statistically significant relationship between the field of study and experiencing toothache (*p* = 0.000). Although this type of pain is present among students of all majors, dental students have experienced it less often (57.3%) than both medical and pharmacy students (79.7%). There was no difference in the choice of over-the-counter painkillers in cases of odontogenic pain between students of dentistry and both medicine and pharmacy (*p* = 0.269). In turn, the choice of nimesulide was very frequent in the group of dental students (40.0%) compared to students of medicine and pharmacy (40.8%), and was similar to the choice of ketaprofen in terms of frequency (40.0%) (*p* = 0.000).

### 3.4. Correlation between Gender and Taking Painkillers among Students of Medicine, Dentistry, and Pharmacy

There was a statistically significant relationship between female gender and the frequency of using analgesics (*p* = 0.032). It was noted that a large percentage of female pharmacy students used painkillers less than once a month compared to female students of medicine or dentistry. However, there was no difference in the pain threshold before taking painkillers between female and male students (*p* = 0.322). Female medical students, more often than other students, did not comply with the recommended doses. There were statistically significant differences in the answers to this question for female gender (*p* = 0.020). Moreover, it was noted that female gender determined concerns about painkillers’ side effects (*p* = 0.030). On the other hand, neither female nor male gender reported being susceptible to drug advertisements. Females experienced odontogenic pain more frequently (*p* = 0.003), but this did not affect the choice of over-the-counter pain medication in the event of such pain (*p* = 0.434). In turn, female gender determined the choice of a prescription drug in cases of odontogenic pain (*p* = 0.016). Similar results related to the choice of prescription drug in cases of odontogenic pain were revealed for male students (*p* = 0.013).

## 4. Discussion

Students of various medical faculties gaining knowledge in pharmacology during their studies constitute a specific research group. Their preferences in basic drug choice may be modified by their acquiring specific professional competences and may differ from the preferences characteristic of the same age group if not involved in medical studies. An assessment of these preferences during their studies is important for the future decision-making process regarding the choice of analgesics. In future, current students of medicine, dentistry, and pharmacy will shape drug policy, recommend that patients use individual drugs, or perform important advisory functions. Therefore, we made an attempt to examine preferences in the choice of painkillers among students of medicine, dentistry, and pharmacy.

The frequency of painkiller use in our study was similar to the results obtained by Duyster et al., who surveyed such use among school-leavers and university students aged 17–25 years in Australia [5]. The vast majority of that study’s participants took painkillers once a month or less than once a month. This is consistent with our research and indicates that people of a similar age who are generally healthy and do not experience chronic pain use analgesics occasionally, with a similar frequency. The paper also showed a relationship between gender and the frequency of medication application. Similar to the results obtained in other studies, women are more likely to use painkillers [5]. This may be due to women being more susceptible to migraines and pain associated with their menstrual cycle, indicating a strong relationship between gonadal steroid hormones and basal pain responses [6,7,8]. Moreover, a relatively high percentage of males among the medicine students in our research showed statistically less frequent use of analgesics compared to dental and pharmacy students. The statistical differences obtained in our research among students of different fields are more likely to be attributable to the differing sex distribution in these study groups.

The majority of surveyed students, regardless of their field of study, reach for painkillers in cases of severe pain. This is in contrast to other studies conducted in a similar age group, where almost half of the respondents use painkillers with the first pain symptoms [5]. In the same study, it was noted that the percentage of people choosing causal treatment instead of painkillers increased with age [5]. Moreover, it is worth noting that, in our study, few respondents declared that they take painkillers preventively, before expected pain. This is in contrast with other studies, where this policy of painkiller use is closely linked to depression, anxiety or sleep disorders [9,10]. People without medical competence very often use available painkillers in such cases. A high level of autonomy resulting from education and extensive knowledge of the effects of painkillers results in a very conscious use of painkillers, limited only to severe pain.

In our research, students, regardless of their field of study, adhered to the recommended doses of drugs. These results are consistent with the data from other studies covering a similar age group, in which respondents denied there being greater effectiveness of painkillers when taken in an increased dose. [5,11]. This is due to a generally low confidence in the safety of analgesics, as reported in earlier studies. The respondents are not convinced that the use painkillers is 100% safe. This belief is independent of gender, level of education, and affiliation with a medical profession [5]. The opposite results regarding analgesics safety were obtained in a survey conducted by Wiliński et al. among Polish students of various fields of study, where 52% of the respondents considered painkillers to be safe, causing side effects only occasionally [11]. It was believed by 65.2% of the respondents that the risk of side effects grows with an increased dose or with the prolonged application of a drug [11]. Other answers obtained in our survey showed that students of medicine, dentistry, and pharmacy use painkillers in a rational way, consistent with medical knowledge. They apply them only in justified cases, in accordance with the recommendations. The most important reason affecting their decisions is the direct cause and effect relationship, i.e., pain and the need to relieve it quickly and effectively. Unlike other survey respondents, non-medical factors such as the drug’s availability or price have little impact on their choice [5]. A drug’s fast onset of action, as well as its effectiveness and potency, are more important than the safety of its use and the lack of serious side effects. Seemingly, a slight correlation between possible side effects and the choice of an individual painkiller results from the fact that our respondents were healthy young people with few contraindications to the application of analgesics. Moreover, the possible side effects of analgesics had a relatively small impact on the choice made by pharmacy students, in comparison to dental and medical students, which may be explained by their limited practical experience in diagnosing and treating the side effects of drugs. During their studies, pharmacy students do not have an opportunity to clinically observe the side effects of currently used drugs.

Ibuprofen was found to be the most preferred over-the-counter pain-reliever for acute pain among all students participating in the study. These results differ from earlier surveys conducted among Polish students, where the majority of respondents indicated paracetamol as the first-line drug for most medical conditions associated with pain [11]. In the same studies, paracetamol was also more frequently used to treat initial symptoms of a cold or sore throat, although it shows a weaker anti-inflammatory effect. Theoretically, this should be less effective than non-steroidal anti-inflammatory drugs (NSAIDs) [11]. NSAIDs were preferred for post-traumatic pain and muscle pain [11]. Both drugs are easily available. Still, our respondents chose ibuprofen more readily, as they presented greater awareness of its stronger anti-inflammatory effect, and therefore more consciously intended its use to eliminate inflammation, a likely cause of pain. A widespread use of NSAIDs is also reflected in cases of gastrointestinal disorders, due to the most well-known side effect of their use. This is consistent with a previous study where patients indicated gastrointestinal disorders as the main side-effect of NSAIDs [12]. In turn, the correlation between gastrointestinal disorders and analgesics’ application was more frequent among students of older years than those of earlier years, who indicated a headache to be the main side effect of analgesics. It is important to emphasize that the choice of analgesics was more diverse among pharmacy students. Ibuprofen was still indicated as the first-line drug for acute pain, but the choices were more diversified. This may be indicative of a better knowledge of the drug’s availability and the less schematic therapy choices among pharmacy students.

Another issue examined in our survey was whether TV commercials can influence the choice of drugs. We compared our results to other studies conducted among the Polish population, subjected to the same information pressure [4,13]. Medicines are one of the most advertised products in Poland. This policy can lead to the excessive use of medicinal products or to addiction. In the group we studied, this potential impact was small and denied by the majority of respondents. This was probably related to the high degree of acquired professional competences and knowledge, which is not available to other potential respondents from the same age group. A possible great impact of television commercials on the choice of widely available analgesics is evidenced by the study conducted by Buczak et al. among high school students in Poland, which showed that 75% of students can point to at least four specific commercials advertising painkillers [4]. Moreover, 47.1% of the students reached for drugs they know from commercials, and 94.6% of them were convinced that using painkillers is socially accepted. [4]. On the other hand, Radlińska et al. obtained contrary results and reported that 68.5% of the respondents declared advertisements to have no effect on their decision to purchase a specific over-the-counter drug [13]. Although these studies denied the direct impact of advertising on the purchase of over-the-counter drugs, it was noted that advertisements misinformed respondents about the properties of drugs and their side effects, and contributed to respondents’ incorrect assessment of their own health [13]. Moreover, elderly people have been found to be more susceptible to this misinformation [13]. In our research, it seems that a young age and high level of education, as well as acquired professional competences, caused the influence of advertisements to be limited or denied.

Students of all studied faculties indicated their studies’ curricula to be their main source of information about painkillers. This shows how strong a determinant of their choice is their acquired professional competence. In addition, the differences between students in the initial and final years of studies indicate that, with the development of professional competence, this factor’s participation in individual choices increases. The results show that the development of autonomy and professional competences acquired in the course of studies increases with age. Contrary to previous studies conducted in corresponding age groups, where the Internet and the information obtained from parents were indicated to be the primary source of information, the influence of the studies’ content was dominant in our research [14,15]. Although other studies revealed that young adults are less likely to use leaflets compared to older ones, in our research, this percentage was relatively high. This results from a better understanding of the specific medical language used among students of medical universities [16,17,18]. Wiliński et al. found that students of medicine, pharmacy, emergency medicine, and nursing more often declared that their custom was to always read medication leaflets, compared to students of other fields of study [11]. Students of dentistry less often indicated studies as their primary source of information on painkillers compared to students of medicine and pharmacy. This may be related to their different study program, which emphasises dental practice over theoretical pharmaceutical knowledge.

Another aim of the study was to assess preferences regarding the use of analgesics to relieve toothache. Pain originating from the teeth has a rich symptomatology and various causes. Frequently, it may involve the surrounding tissues and be non-odontogenic in nature. This group includes pain in the teeth and surrounding tissues, pain caused by injuries to the teeth and maxillofacial area, and pain associated with treatment [19]. In many cases, toothache can be relieved by implementing an effective causal endodontic, conservative or surgical treatment. Negligence may intensify the symptoms of inflammation, including pain, and lead to serious complications. Although the possible causes of odontogenic pain and the options of causal treatment to reduce toothache are known to dentistry students, they did not affect the choice of analgesics to fight odontogenic pain. In addition, shaping preferences in the choice of analgesics is important for pharmacists and general practitioners. Although they are not able to remove the cause of pain, they face the need to implement an effective pharmacological therapy or advise patients in choosing over-the-counter painkillers [19]. Our results confirmed that students of all faculties have recommendations concerning the painkillers that could be applied to relieve toothache and implement them in their daily practice. It is this knowledge that determines their preferences. The choice of ibuprofen agrees with general guidelines, which recommend its use, either together with paracetamol or alone, as the first choice for odontogenic pain. NSAIDs have an advantage over opioid drugs, which produce a weaker analgesic effect and more side effects, and are used at the lowest possible doses and for a short time. Moreover, paracetamol monotherapy is only used if there are contraindications to NSAIDs. It is more often used together with ibuprofen, which increases the analgesic effect [19]. The same recommendations apply to combating pain resulting from surgical removal of the lower molars, including the third molars, as well as pain following pulpotomy in primary molars [20,21]. Previous studies showed that, for pain caused by oral surgery, 400 mg of ibuprofen is more effective than 600 mg acetaminophen combined with 60 mg codeine, and as effective as the use of 650 mg acetaminophen combined with 10 mg of oxycodone [22]. Ibuprofen, as an NSAID, should be treated as “the first-line analgesic” for acute postprocedural dental pain [19,20,21,22,23]. Moreover, it is effective in controlling the most common odontogenic pain caused by pulpitis and periodontitis [24]. In addition, the choice of a painkiller for toothache may be influenced by its previous effectiveness. Students of dentistry turned out to experience pain of dental origin, probably due to their better ability to recognise dental problems as compared to students of medicine and pharmacy. In turn, the choice of nimesulide as a prescription drug for toothache among students of dentistry proves their good knowledge of pharmacological therapy for selected diseases of the head and neck, where the use of nimesulide to treat temporomandibular joint pain, osteoarthritis, and cranial nerves’ neuralgia is recommended [25]. Previous studies have revealed nimesulide to be the second most frequently prescribed painkiller, after ibuprofen, in dental outpatients [26]. Moreover, nimesulide is widely used to fight post-extraction pain [27]. Bocanegra et al. reported that nimesulide and ibuprofen provided similarly effective 24 h pain relief after the removal of an impacted third molar. However, the analgesic effect of nimesulide was faster and stronger than that of ibuprofen [28]. In turn, the wide use of ketoprofen and nimesulide in the treatment of dental pain in all students is consistent with the observation that both drugs are equally effective in controlling pain and inflammation after the surgical removal of the lower wisdom teeth [25]. Among medical and pharmacy students, ketoprofen is an anti-inflammatory drug, which is recommended for toothache and preferred over nimesulide. This agrees with previous studies, which have shown that ketoprofen provides a faster analgesic and anti-inflammatory effect than ibuprofen, which is desirable in dental pain [29]. Ketoprofen is especially recommended after surgical removal of the lower wisdom tooth because, while ensuring a good analgesic and anti-inflammatory effect, it does not change the haematological parameters that are important in a surgical procedure, such as prothrombin time, activated partial thromboplastin time, clot retraction, and platelet count [30]. Students of earlier years believe that pain of dental origin does not require a prescription drug. This may result from their limited clinical and pharmacology knowledge. A significant number of respondents do not prefer opioids such as tramadol in cases of dental pain. Tramadol is more commonly used in these cases, in combination with paracetamol or a non-steroidal anti-inflammatory drug, showing a faster analgesic effect in combination with a non-steroidal anti-inflammatory drug such as dexketoprofen [31,32]. In turn, the rare use of coxibs to control dental pain may be associated with an increased risk of myocardial infarction and cardiovascular complications following their use, as often reported in studies. Their use in cases of mild to moderate pain, which is often associated with dental causes, may be risky, as these complications may be expected [33].

The limitations of the study are due to there being no available data related to previous pain experience among responders. Previous experience of pain and medical history could impact the current perception of pain and choice of painkillers. Another limitation of the study is the lack of familial history and a lack of information about other possible sources of knowledge on painkillers. Family members, their medical histories, and the level of medical education before their studies might shape individual preferences in taking painkillers. The questionnaire used in our study contained items assessing preferences, resulting in low α Cronbach value. We can treat our research as a pilot study to build new, valuable, and better validated tools in the future.

## 5. Conclusions

Medical, dentistry, and pharmacy studies determine the personal preferences of students. The progress of studies and their own specificity differently shape personal choices regarding drugs and students’ perception of their safety, indications, and precautions. They provide resistance to marketing and advertising, and provide greater autonomy when choosing the drugs that are to be used. On the basis on the results obtained in our study, we can indicate that the knowledge acquired during studies and their specific character determine the choice of painkillers, low susceptibility to advertisements, and the use of painkillers, in accordance with their pharmacological recommendations. The specificity of studies, gender, and year of study are the most important determinants of the choice of painkillers. Small differences in preferences regarding the choice of painkillers, and the perception of pain and its management between student groups, proves that future policy-making in this area requires the simultaneous and equal participation of all groups of future medical professionals included in the study.

## Figures and Tables

**Figure 1 healthcare-12-00196-f001:**
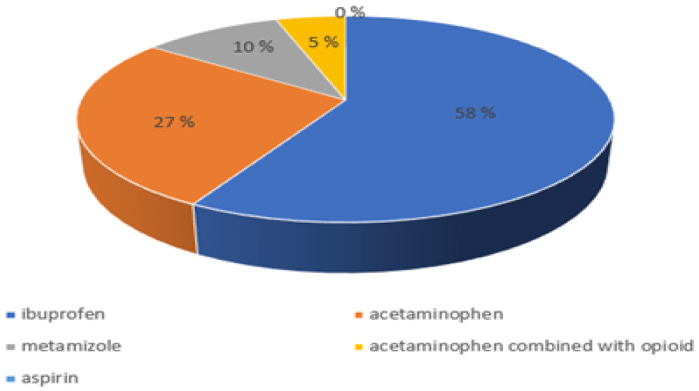
Distribution of preferences in the choice of over-the-counter painkillers for acute pain among all students.

**Figure 2 healthcare-12-00196-f002:**
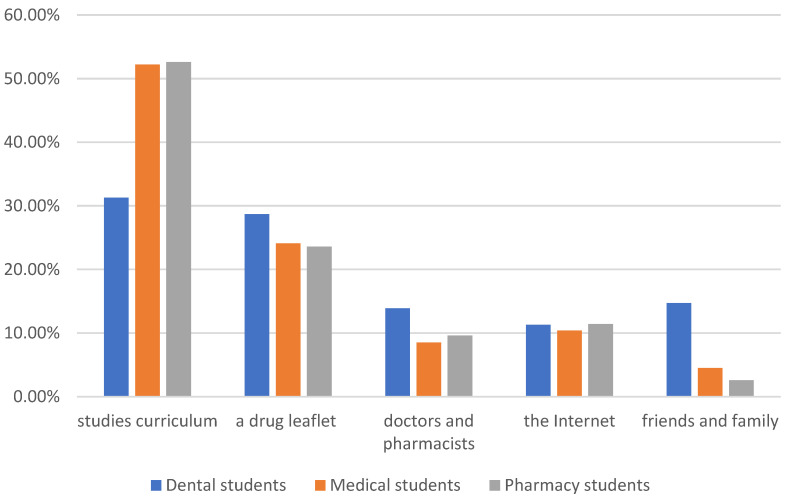
Gaining information on painkillers among students of medicine, dentistry, and pharmacy.

**Figure 3 healthcare-12-00196-f003:**
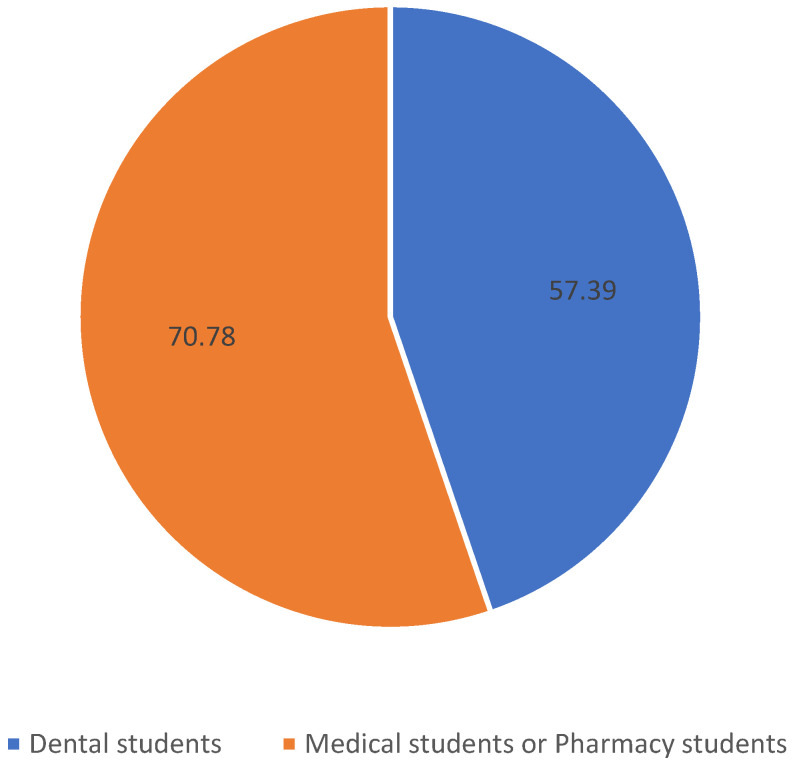
Comparison of experience of toothache among medical students, dental students, and pharmacy students.

**Table 1 healthcare-12-00196-t001:** Demographic characteristics and distribution of answers of students of medicine, dentistry, and pharmacy.

Parameters	All Students	Medicine Students	Dental Students	Pharmacy Students	*p* Value
Number of individuals, n (%)	382	153 (40.0)	115 (30.1)	114 (29.8)	
Gender:					**0.000 ^a^**
Female, n (%)	303 (79.1)	106 (34.9)	97 (32.0)	100 (33.0)	
Male, n (%)	79 (20.6)	47 (59.4)	18 (22.7)	14 (17.7)	
**Frequency of painkillers application, n (%)**					**0.034 ^b^**
More than once a week	15 (3.9)	11 (7.1)	2 (1.7)	2 (1.7)	
Once a week	59 (15.4)	21 (13.7)	24 (20.8)	14 (12.2)	
Once a month	180 (47.1)	63 (41.1)	58 (50.4)	59 (51.7)	
Less than once a month	128 (33.5)	58 (37.9)	31 (26.9)	39 (34.2)	
**Pain threshold at which you use a painkiller, n (%):**					0.547 ^b^
To prevent expected pain	3 (0.7)	0 (0.0)	2 (1.7)	1 (0.8)	
Mild pain	3 (0.7)	2 (1.3)	0 (0.0)	1 (0.8)	
Moderate pain	109 (28.5)	45 (29.4)	32 (27.8)	32 (28.0)	
Severe pain	182 (47.6)	77 (50.3)	56 (48.7)	49 (42.9)	
Unbearable pain	85 (22.2)	29 (18.9)	25 (21.7)	31 (27.1)	
**Do you comply with recommended doses of a drug, n (%)?**					0.152 ^b^
No	2 (0.5)	1 (0.6)	1 (0.8)	0 (0.0)	
Rather no	9 (2.3)	7 (4.5)	1 (0.8)	1 (0.8)	
Rather yes	88 (23.0)	30 (19.6)	33 (28.7)	25 (21.9)	
Yes	283 (74.0)	115 (75.1)	80 (69.5)	88 (77.1)	
**What is the most important for you when choosing a painkiller, n (%)?**					**0.008 ^b^**
Fast onset of action	179 (46.8)	60 (39.2)	68 (59.1)	51 (44.7)	
Drug potency	125 (32.7)	52 (33.9)	26 (22.6)	47 (41.2)	
no side effects	22 (5.7)	11 (7.1)	7 (6.0)	4 (3.5)	
duration of analgesic effect	30 (7.8)	19 (12.4)	5 (4.3)	6 (5.2)	
drug availability	20 (5.2)	10 (6.5)	7 (6.0)	3 (2.6)	
price	6 (1.5)	1 (0.6)	2 (1.7)	3 (2.6)	
**Do you have contraindications to the use of painkillers, n (%)?**					0.845 ^a^
no	359 (93.9)	144 (94.1)	109 (94.7)	106 (92.9)	
yes	23 (6.0)	9 (5.8)	6 (5.2)	8 (7.0)	
**Are you concerned about side effects when using mild painkillers, n (%)?**					**0.047 ^a^**
no	284 (74.3)	111 (72.5)	79 (68.7)	94 (82.4)	
yes	98 (25.6)	42 (27.4)	36 (31.3)	20 (17.5)	
**Which drug do you use in case of acute pain, n (%)?**					**0.027 ^b^**
ibuprofen	221 (57.8)	77 (50.3)	75 (65.2)	69 (60.5)	
acetaminophen	102 (26.7)	49 (32.0)	29 (25.2)	24 (21.0)	
metamizole	37 (9.6)	21 (13.7)	5 (4.3)	11 (13.7)	
acetaminophen combined with opioid	21 (5.4)	6 (3.9)	6 (5.2)	9 (7.8)	
aspirin	1 (0.2)	0 (0.0)	0 (0.0)	1 (0.8)	
**Do you consider advertisements when choosing painkillers, n (%)?**					0.787 ^b^ 0.924 ^b^
no, never	353 (92.4)	143 (93.4)	106 (92.1)	104 (91.2)	
yes:	29 (7.5)	10 (6.5)	9 (7.8)	10 (8.7)	
sometimes	25 (6.5)	9 (5.8)	7 (6.0)	9 (7.8)	
often	3 (0.7)	1 (0.6)	1 (0.8)	1 (0.8)	
always	1 (0.2)	0 (0.0)	1 (0.8)	1 (0.8)	
**Where do you get information about individual painkillers from, n (%)?**					**0.001 ^a^**
studies curriculum	176 (46.0)	80 (52.2)	36 (31.3)	60 (52.6)	
drug leaflet	97 (25.3)	37 (24.1)	33 (28.7)	27 (23.6)	
doctors and pharmacists	40 (10.4)	13 (8.5)	16 (13.9)	11 (9.6)	
the Internet,	42 (10.9)	16 (10.4)	13 (11.3)	13 (11.4)	
friends and family	27 (7.0)	7 (4.5)	17 (14.7)	3 (2.6)	
**What is the most common side effect of painkillers, n (%)?**					0.741 ^b^
gastrointestinal complaints,	333 (87.1)	138 (90.2)	97 (84.3)	98 (85.9)	
headaches and dizziness,	30 (7.8)	9 (5.8)	11 (9.5)	10 (8.7)	
urticaria,	12 (3.1)	3 (1.9)	5 (4.3)	4 (3.5)	
anaphylactic shock,	4 (1.0)	1 (0.6)	1 (0.8)	2 (1.7)	
excessive sweating	3 (0.7)	2 (1.3)	1 (0.8)	0 (0.0)	
**Have you experienced toothache, n (%)?**					**0.000 ^a^**
yes	279 (73.0)	122 (79.7)	66 (57.3)	91 (79.8)	
no	103 (26.9)	31 (20.2)	49 (42.6)	23 (20.1)	
**What over-the-counter medicine would you recommend for toothache, n (%)?**					0.080 ^b^
Ibuprofen	213 (55.7)	73 (47.7)	70 (60.8)	70 (61.4)	
Acetaminophen	88 (23.0)	42 (27.4)	27 (23.4)	19 (16.6)	
No OTC drug. A prescription drug would be my first choice	30 (7.8)	15 (9.8)	5 (4.3)	10 (8.7)	
I would not recommend any OTC drug	22 (5.7)	9 (5.8)	8 (6.9)	5 (4.3)	
Metamizole,	25 (6.5)	14 (9.1)	4 (3.4)	7 (6.1)	
Aspirin	4 (1.0)	0 (0.0)	1 (0.8)	3 (2.6)	
**What prescription drug would you recommend for dental pain, n (%)?**					**0.003 ^b^**
Ketoprofen	189 (49.4)	85 (55.5)	47 (40.8)	57 (50.0)	
Nimesulide	104 (27.2)	27 (17.6)	46 (40.0)	31 (27.1)	
Tramadol	31 (8.1)	16 (10.4)	5 (4.3)	10 (8.7)	
I wouldn’t choose a prescription drug	49 (12.8)	19 (12.4)	17 (14.7)	13 (11.4)	
Celecoxib	9 (2.3)	6 (3.9)	0 (0.0)	3 (2.6)	

^a^, Chi2 Pearson’s test; ^b^, Fisher-Freeman-Halton’s test; OTC, over-the-counter; bold values refer to statistically significant ones.

**Table 2 healthcare-12-00196-t002:** Correlations between the year of study and the use of analgesics.

Parameters	Students of ≥4th Year	Students of ≤3rd Year	*p* Value
**Frequency of painkillers application:**			**0.016 ^a^**
More than once a week	13 (6.0)	2 (1.2)	
Once a week	26 (12.0)	33 (19.7)	
Once a month	107 (49.7)	73 (43.7)	
Less than once a month	69 (32.0)	59 (35.3)	
**Pain threshold at which you use a painkiller:**			0.087 ^b^
To prevent expected pain	2 (0.9)	1 (0.6)	
Mild pain	2 (0.9)	1 (0.6)	
Moderate pain	61 (28.3)	48 (28.7)	
Severe pain	106 (49.3)	76 (45.5)	
Unbearable pain	44 (20.4)	41 (24.5)	
**Do you comply with recommended doses of a drug?**			0.697 ^b^
No	2 (0.9)	0 (0.0)	
Rather not	4 (1.8)	5 (2.9)	
Rather yes	50 (23.2)	38 (22.7)	
Yes	159 (73.9)	124 (74.2)	
**What is the most important for you when choosing a painkiller?**			0.161 ^b^
Fast onset of action	98 (45.5)	81 (48.5)	
analgesic drug potency	78 (36.2)	47 (28.1)	
no side-effects	15 (6.9)	7 (6.5)	
duration of analgesic effect	14 (6.5)	16 (9.5)	
drug availability	8 (3.7)	12 (7.1)	
price	2 (0.9)	4 (2.4)	
**Do you have contraindications to the use of painkillers?**			0.185 ^a^
no	199 (92.5)	160 (95.8)	
yes	16 (7.4)	7 (4.1)	
**Are you concerned about side effects when using mild painkillers?**			0.501 ^a^
No	157 (73.0)	127 (76.0)	
Yes	58 (26.9)	40 (23.9)	
**What drug do you use in case of acute pain?**			0.156 ^b^
ibuprofen	129 (60.0)	92 (55.0)	
acetaminophen	51 (23.7)	51 (30.5)	
metamizole	25 (11.6)	12 (7.1)	
acetaminophen combined with opioid	9 (4.1)	12 (7.1)	
aspirin	1 (0.4)	0 (0.0)	
**Do you consider advertisement when choosing painkillers?**			
no, never	202 (93.9)	151 (90.4)	0.443 ^b^
yes	13 (6.0)	16 (9.5)	0.195 ^b^
sometimes	12 (5.5)	13 (7.7)	
often	1 (0.4)	2 (1.2)	
always	0 (0.0)	1 (0.6)	
**Where do you get information about individual painkillers from?**			**0.000 ^a^**
studies curriculum	119 (55.3)	57 (34.1)	
drug leaflet	50 (23.2)	47 (28.1)	
doctors and pharmacists	16 (7.4)	24 (14.3)	
the Internet	24 (11.1)	18 (10.7)	
friends and family	6 (2.7)	21 (12.5)	
**What is the most common side effect of painkillers?**			**0.000 ^a^**
gastrointestinal complaints	197 (91.6)	136 (81.4)	
headaches and dizziness	6 (2.7)	24 (14.3)	
urticaria	10 (4.6)	2 (1.2)	
anaphylactic shock	0 (0.0)	4 (2.4)	
excessive sweating	2 (0.9)	1 (0.6)	
**Have you experienced toothache?**			0.102 ^a^
yes	150 (69.7)	129 (77.2)	
no	65 (30.2)	38 (22.7)	
**What over-the-counter medicine would you recommend for toothache?**			**0.030 ^b^**
Ibuprofen	128 (59.5)	85 (50.9)	
Acetaminophen	48 (22.3)	40 (23.9)	
No OTC drugs, a prescription drug would be my first choice	16 (7.4)	14 (8.3)	
I would not recommend any	6 (2.7)	16 (9.5)	
Metamizole,	13 (6.0)	12 (7.1)	
Aspirin	4 (1.8)	0 (0.0)	
**What prescription drug would you recommend for dental pain?**			**0.000 ^b^**
Ketoprofen,	105 (48.8)	84 (50.3)	
Nimesulide,	75 (34.8)	29 (17.3)	
Tramadol,	14 (6.5)	17 (10.1)	
I wouldn’t choose a prescription drug,	16 (7.4)	33 (19.7)	
Celecoxib	5 (2.3)	4 (2.4)	

^a^, Chi2 Pearson’s test; ^b^, Fisher–Freeman–Halton’s test; OTC, over- the -counter; bold values refer to statistically significant ones.

## Data Availability

The data presented in this study are available on request from the corresponding author.

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
