# Peer review of "Assessment of Preferences in Taking Painkillers among Students of Medicine, Dentistry, and Pharmacy: A Pilot Study"

_healthcare, 2024, doi:10.3390/healthcare12020196_

Round 1

Reviewer 1 Report (Previous Reviewer 3)

Comments and Suggestions for Authors

Thank you for allowing me to review this cross-sectional study that assessed student attitudes about painkillers.

The topic is interesting, but the implementation requires corrections:

1. Abstract: "Fast onset of an analgesic action was most important for medical (39.22%) and dental students (59.1%), compared to pharmacy students (44.7%)." - according to these percentages, this is more important for pharmacists than for medics.

2. Methodology - who participated in the design of the questionnaire, and on the basis of which it was designed (which books or scientific papers) and whether it passed the pilot study and validation.

3. Results: Is there a possibility of dividing table 1 into several parts because it is so illegible and difficult to follow.

Author Response

Thank you very much for your comments and suggestions. In response to your review please find our answers below:

Ad. Abstract:

Percentage values have been corrected.

Ad. Methodology:

 Questions contained in the survey have been formulated after the initial interviews with medicine, dentistry and pharmacy students representatives. We tried to collect and include all suggestions reported in the interviews. All the authors participated in preparing final questionnaire design. We searched literature for papers concerning research design. Some of them have been attached as references but in fact we cited these items more to support or interpret the obtained results in the discussion subsection rather than to support methodology of our study. This is because our aim was not to simply transmit questions from other surveys.  We intended to prepare  a new, our own and self-made survey, which in future will be used to build even more accurate questionnaires aimed at studying selected issues which would appear worth closer examination.  We did not use any books in preparing the survey. Our questionnaire has not got any references to other research tools of that kind. In the vast majority our questionnaire contains items without gradation. It is quite common in similar kind of surveys studying preferences. For validation purposes we have checked α Cronbach value for the whole questionnaire and it was 0.32 (including items with gradation, only). Our questionnaire containing many items without gradation caused the Cronbach value to be very low and hardly accepted in validation purposes. Therefore, we can consider our questionnaire to be the starting point to build better validated and more valuable research tools. We can treat our project as a pilot study. Resultingly, the term “pilot study“ was included in the title. 

Ad. Results:

The results presented in Table 1 have been consolidated to one table in response to the reviewer’s recommendations suggested in the previous round of reviewing process. The current versions of the manuscript and table 1 follow these suggestions. In the previous version table 1 presented the summarized answers to questions included in the questionnaire without division into assessed faculties. Table 2 was a comparison of demographic data and answers to questions between the students of medicine, pharmacy and dentistry. Moreover, additional figures and pie charts related to the chosen answers from the survey have been introduced to the manuscript in response to the reviewer’s suggestions. The authors made their best to implement all suggested improvements to the  presentation and to make the manuscript more readable. However, sometimes the postulated changes seemed conflicting in the subsequent rounds of the reviewing process making them hard to incorporate in the paper. 

Reviewer 2 Report (Previous Reviewer 4)

Comments and Suggestions for Authors

No additional comments. Article has been improved significantly than it was before.

Author Response

Thank you very much for your review

Reviewer 3 Report (New Reviewer)

Comments and Suggestions for Authors

I thoroughly read the manuscript entitled Assessment of Preferences in Taking Painkillers among Students of Medicine, Dentistry, and Pharmacy – a Pilot Study

Although the manuscript contains all the necessary elements (IMRAD), the data from only one area, which is already well known, with an overly extensive presentation of the results (especially descriptive statistics) do not have sufficient scientific impact.

Also, it is not clear why this is a pilot study. What major project/research follows up on the results obtained?

Author Response

Thank you very much for your review.

We tried to present all obtained results comprehensively. Except of descriptive statistical analysis the detailed comparison between assessed study groups with additional statistical tests addressed to these purpose has been conducted.

We hope that our study will prove interesting, useful for many recipients and able to make a scientific sound.

We intended to prepare  a new, our own and self-made survey, which in future will be used to build even more accurate questionnaires aimed at studying selected issues which would appear worth closer examination. Therefore, we can consider our questionnaire to be the starting point to build more valuable and more specified  research tools. We can treat our project as a pilot study. It seems that some issues included into survey are worth deeply analysing and broadening.

Round 2

Reviewer 1 Report (Previous Reviewer 3)

Comments and Suggestions for Authors

Congrats authors. Now manuscript is much better. But there is still question of validation of survey. 

Author Response

Thank you very much for your review.

Our questionnaire has not got any references to other research tools of that kind. In the vast majority our questionnaire contains items without gradation. It is quite common in similar kind of surveys studying preferences. For validation purposes we have checked α Cronbach value for the whole questionnaire and it was 0.32 (including items with gradation, only). Our questionnaire containing many items without gradation caused the Cronbach value to be very low and hardly accepted in validation purposes. Therefore, we can consider our questionnaire to be the starting point to build better validated and more valuable research tools. We can treat our project as a pilot study. Resultingly, the term “pilot study“ was included in the title.  

Reviewer 3 Report (New Reviewer)

Comments and Suggestions for Authors

The manuscript can be accepted without any further changes

Author Response

Thank you vey much for your review

This manuscript is a resubmission of an earlier submission. The following is a list of the peer review reports and author responses from that submission.

Round 1

Reviewer 1 Report

Comments and Suggestions for Authors

The article presents an interesting study that addresses preferences in the choice of analgesics among medical, dental, and pharmacy students. The study highlights the importance of understanding these preferences as current students in these disciplines will play a crucial role in future drug policy and in recommending treatments to patients. The relationship between the choice of analgesics and factors such as gender and the knowledge acquired during education is a significant aspect explored.

The article provides a detailed insight into the preferences of students regarding the types of analgesics they choose and how these preferences may be influenced by the acquisition of professional knowledge. Additionally, the impact of advertising on analgesic choice is discussed, adding an interesting element to the study.

The results of the study suggest that medical, dental, and pharmacy students tend to make informed and education-consistent decisions regarding the choice of analgesics. This is a positive aspect and demonstrates the importance of education in health-related decision-making.

I would be interested to know if the authors considered including students from other disciplines, as this might provide a broader perspective on analgesic preferences. Additionally, it would be valuable to understand if the authors believe that the findings can be extrapolated to other countries with different pharmacological policies. It might be worth exploring whether the duration of the interviews was adequate, as extending the interview time could potentially allow for the recruitment of a larger number of participants, leading to a more comprehensive study.

Author Response

Thank you very much for your comments and suggestions. In response to your review please find our answers below.

Ad 1. We did not consider including students from other disciplines to broaden the perspective on the assessment of analgesics preferences. In our opinion students from other non-medical disciplines could confound design of the research. Moreover, different level of medical education and competences in medical and non-medical disciplines may interfere with the final results. In the future we consider comparing the preferences in taking painkillers of medical and non-medical students.

Ad 2. In our opinion the obtained findings should not be extrapolated to other countries with different pharmacological policies as studies curricula vary. The authors do not know the specificity of the studies curricula in other countries and in our opinion the lack of knowledge in this respect makes it impossible to implement our findings.

Ad 3. In our opinion extending the interview time did not increase the effectiveness of the conducted study. The vast majority of the questionnaires responses came from the initial time of the research.

Reviewer 2 Report

Comments and Suggestions for Authors

The study seems to be original and  interesting, however some considerations are required:

-no mention of what kind of study design was carried out,

-no evidence of the blank questionnaire provided,

-descriptive statistics is fine but all the elaboration of the different tests is missing,

-it looks like there is a correlation between the fields of study but it is not clear which courses are compared for each question,

-more stat graphs would be likely to add a glance of the results in a more immediate way,

-English editing needs a major revision as sometimes not properly smooth.

Comments on the Quality of English Language

The paper needs major revision regarding the following sections:

1. Intro (not enough and adequate literature provided),

2. Methods: it needs further details re the study design, the questionnaire and the stat

3. Results & Discussion: more details on stat apart from descriptive. Ideally correlation between the different data using 2 groups comparison.

4. English writing improvement and proofreading.

Author Response

Thank you very much for your comments and suggestions. In response to your review please find our answers below.

Ad 1. We presented the study design in the Material and Methods subsection. Below please find our extended explanation of the issue:

The questions contained in the survey have been prepared after the initial interview with the student representatives from the medicine, dentistry and pharmacy faculties. We tried to collect and include all suggestions reported in the interviews. The study group of respondents has been set on the basis on the total number of students of the medicine, dentistry and pharmacy faculties originating from all Polish medical universities. The designed survey has been addressed and distributed to all Polish students of medicine, dentistry and pharmacy. Based on the data from the Central Statistical Office in Poland, we determined that in the academic year 2022/2023, 37,770 medical students, 8,020 pharmacy students and 5,310 dentistry students studied in Poland. To set the needed number of respondents in each study group of students we used the following values: d=0.10 (an absolute precision, (margin of error)), α= 0.05, N (population size) and thus we received required number of students from medicine, pharmacy and dentistry faculties for the study to be valuable:

Medical students N=96

Dental students N=94

Pharmacy students N=95

Finally, we received responses from the students from Warsaw University of Medical Sciences, Pomeranian University of Medical Sciences in Szczecin, Poznan University of Medical Sciences and University of Medical Sciences in Wroclaw. Other Polish medical universities were not represented.

Ad 2. The whole contents of our questionnaire (blank questionnaire) has been attached in the supplementary material).

Ad 3 and d 4. The statistical tests used to compare selected student groups have been elaborated and broadly presented. In the tables 2 and 3, the specific test types have been assigned to the p values and presented in the upper index. Moreover, the post hoc tests for multiple comparisons with Benjamini-Hochberg method have been used to compare medical students and dental students, medical students and pharmacy students, and dental students and pharmacy students as well as students ≤3rd year of the study and ≥4th year of the study.

Ad 4. More detailed statistical tests for better comparison between students groups have been used.

Ad 5. New stat graphs for better and immediate results assessment have been introduced.

Ad 6. The whole manuscript has been edited by a native speaker twice.

Comments to Editors

Ad 1. More adequate literature has been introduced in the introductory subsection.

Ad 2. More details related to the study design, the questionnaire and the statistics have been added.

Ad 3. More detailed statistical tests for better comparison between two students groups have been used.

Ad 4. The whole manuscript has been edited by a native speaker twice.

Reviewer 3 Report

Comments and Suggestions for Authors

I have identified several questionable points that require correction:

Abstract - Research Methodology and Organization: How was it conducted?

Abstract - Results:

"Ibuprofen was the most frequently preferred over-the-counter painkiller, regardless of the study major." Please provide the specific percentage.

"Pharmacy students expressed the least concern about possible side effects of analgesics." Please provide the percentage in comparison to other groups.

"Fast onset of analgesic action was most important for medical and dental students, compared to pharmacy students." Please state the exact percentage.

"A statistically significant number of medicine and pharmacy students indicated their studies as the main source of information about painkillers, compared to dentistry students." Please provide the p-value.

"There are no statistically significant differences in pain severity at which analgesics are used among the surveyed groups of students." Please state the p-value. And so on for all presented results.

Abstract - Conclusion: The conclusion is currently too general. Please restate the conclusion based on the obtained results.

At the end of the Introduction section, please state the research hypothesis.

Methodology: "(male:female ratio 79:303)" - Please provide this information in percentages. Also, clarify which universities are included, their total number of students, and how the sample size was calculated. If the sample size is small given the number of universities, please address this issue. I suggest aligning the methodology with the SROBE guidelines. Please specify who designed the questionnaire, who validated it, and whether a pilot study was conducted. If the methodology is flawed, please describe how to supplement the questionnaire for clarity.

The description of the statistical methods is inadequate.

Table 1: Please clarify the meaning of "study year ≤3rd" and "˃4th". Additionally, present percentages with one decimal place and p-values with three decimal places.

Please discuss the strengths and limitations of this study.

The conclusion is currently too general and may be more appropriate as part of the discussion. The conclusion should be specific to this research and based on the obtained results.

Lastly, please explain why the research does not have the approval of the ethics commission.

Author Response

Thank you very much for your comments and suggestions. In response to your review please find our answers below.

Ad 1. In abstract, the subsection “Research Methodology and Organization” has been extended and described with more details.

Ad 2. In abstract, in “Results” subsection the suggested percentage values and p value have been introduced. Both percentage values and p-values have been completed and introduced to present all obtained results

Ad 3. In abstract, conclusions have been restated making them less general

Ad 4. At the end of the introduction subsection the research hypothesis has been stated

Ad 5. The percentage values for male:female ratio have been introduced.

The questions contained in the survey have been prepared after the initial interview with the student representatives from the medicine, dentistry and pharmacy faculties. We tried to collect and include all suggestions reported in the interviews. The study group of respondents has been set on the basis on the total number of students of the medicine, dentistry and pharmacy faculties originating from all Polish medical universities. The designed survey has been addressed and distributed to all Polish students of medicine, dentistry and pharmacy. Based on the data from the Central Statistical Office in Poland, we determined that in the academic year 2022/2023, 37,770 medical students, 8,020 pharmacy students and 5,310 dentistry students studied in Poland. To set the needed number of respondents in each study group of students we used the following values: d=0.10 (an absolute precision, (margin of error)), α= 0.05, N (population size) and thus we received required number of students from medicine, pharmacy and dentistry faculties for the study to be valuable:

Medical students N=96

Dental students N=94

Pharmacy students N=95

Finally, we received responses from the students from Warsaw University of Medical Sciences, Pomeranian University of Medical Sciences in Szczecin, Poznan University of Medical Sciences and University of Medical Sciences in Wroclaw. Other Polish medical universities were not represented.

Our questionnaire has not any references to other research tools of that kind. We have checked α Cronbach value for the whole questionnaire and it was 0.32 (including only items with gradation). However, our questionnaire contained many items related to preferences without gradation which caused the Cronbach value to be so low and hardly accepted in validation purposes. Therefore, we can consider our questionnaire to be the starting point to build better validated and more valuable research tools. We can treat our project as a pilot study. We included the term “ pilot study “ in the title.

Ad 6. More detailed statistical tests for better comparison between two students groups have been used. The statistical tests used to compare selected student groups have been elaborated and broadly presented. In the tables 2 and 3, the specific test types have been assigned to the p values and presented in the upper index. Moreover, the post hoc tests for multiple comparisons with Benjamini-Hochberg method have been used to compare medical students and dental students, medical students and pharmacy students, and dental students and pharmacy students as well as students ≤3rd year of the study and ≥4th year of the study.

Ad 7. In table 1 the term "study year ≤3rd" and "˃4th" has been replaced by the term "study year ≤3rd" and "≥4th". Moreover, the term "study year ≤3rd" and "≥4th" has been clarified in table 1

Ad 8. The percentage values with one decimal place and p-values with three decimal places have been presented in the whole manuscript.

Ad 9. The limitations of this study have been indicated and inserted into discussion subsection.

Ad 10. We have indicated gender, character of the studies, their progress as main determinants in the choice of painkillers. We have tried to present conclusions more clearly.

Ad 10. According to the recommendations of the Bioethics Committee of the Poznan University of Medical Sciences, completely anonymous surveys based on own questionnaires and not relating to intimate life and not relating to psychiatry and pediatrics do not require the consent of the committee to conduct them. According to the Polish law and GCP regulations our research does not require approval of any Bioethics Committee. Our research does not possess features of a medical experiment. For the publication purposes the Bioethics Committee at Poznan University of Medical Sciences confirmed that this research is not a medical experiment (decision number KB-887/23).

Reviewer 4 Report

Comments and Suggestions for Authors

The survey article by BÅ‚ochowiak and et. al. summarizes, assess, and compare preferences in taking painkillers by the group of dentals, medical and pharmacy students and to find possible factors determining their preferences. Their study consists of total 382 students from pharmacy, medicine, and dentistry majors. Survey reveals that the Ibuprofen was the most frequently preferred over-the-counter painkiller, regardless of the students from among 3 study majors. Further, the pharmacy, medicine (pharmacy and medicine are majority) and dentistry students indicated their studies to be the main source of information about determining the painkillers. Small differences in preferences regarding the choice of painkillers and the perception of pain and its management between student groups proves that future policy making in this area requires simultaneous and equal participation of all groups of future medical professions in large numbers.  

Overall article is informative and indicative of the preliminary trend, yet the presented stats can be expandable for broader number of participants including additional preferential factors (medical/non-medical) to extract more generalize trend in preferring painkillers or similar drugs in order to generate health policies geographically. As such manuscript is well written, and data is supportive to the conclusions. Therefore, I recommend this manuscript to consider for the publication after minor revision that are highlighted in the attached pdf file.

Author Response

Thank you very much for your comments and suggestions. In response to your review please find our answers below.

Ad 1. Additional figures have been added and introduced into the manuscript

Ad 2 Abstract has been rewritten according to the journal format

Ad 3 Additional factors possibly determining the students choice of painkillers have been formulated, emphasized and introduced in the aim of the study.

Ad 4 The detailed data regarding the age of respondents have been completed. Age of our responders was very homogenous and in our opinion it did not affect the obtained findings.

Ad 5 The type of the statistical tests used for the p-value calculation have been completed in the presented tables

Ad 6 The rows in the tables have been aligned

Round 2

Reviewer 2 Report

Comments and Suggestions for Authors

The manuscript shows the improvements that the Authors have been asked to apply. The methodology has been furtherly clarified and the information are comprehensive now.

The grammar/language editing and proofreading has been done and it seems to be suitable to the publication.

Reviewer 3 Report

Comments and Suggestions for Authors

Thank you to the authors for incorporating the suggested corrections. The manuscript has shown improvement; however, the methodology and interpretation of the results themselves appear to be lacking depth and interest.

Is the questionnaire used in this study validated? If not, could you provide information on its designer, the validation process, and the basis on which it was developed?

Additionally, please include details on the inclusion and exclusion criteria. What criteria were applied, and how were they determined? Could you also provide information on the sample size calculation and the rationale behind it?

What was the time frame for conducting the research?

Observation: Tables 1 and 2 seem similar. Could there be a consolidated table (e.g., an "aggregate table") that includes totals for Medicine, Farmacy, and Dentistry, allowing for a clearer comparison between them?

In my opinion, the manuscript appears suitable for a local journal. However, there are concerns about the questionnaire's design and the presentation of results, which may need further attention and improvement.